

# Targeted NGS for species level phylogenomics: "made to measure" or "one size fits all"?

Malvina Kadlec[1], Dirk U. Bellstedt[2], Nicholas C. Le Maitre[2] and Michael D. Pirie[1]

[1] Institut für Organismische und Molekulare Evolutionsbiologie, Johannes-Gutenberg Universität Mainz, Mainz, Germany
[2] Department of Biochemistry, University of Stellenbosch, Stellenbosch, South Africa

## ABSTRACT

Targeted high-throughput sequencing using hybrid-enrichment offers a promising source of data for inferring multiple, meaningfully resolved, independent gene trees suitable to address challenging phylogenetic problems in species complexes and rapid radiations. The targets in question can either be adopted directly from more or less universal tools, or custom made for particular clades at considerably greater effort. We applied custom made scripts to select sets of homologous sequence markers from transcriptome and WGS data for use in the flowering plant genus *Erica* (Ericaceae). We compared the resulting targets to those that would be selected both using different available tools (Hyb-Seq; MarkerMiner), and when optimising for broader clades of more distantly related taxa (Ericales; eudicots). Approaches comparing more divergent genomes (including MarkerMiner, irrespective of input data) delivered fewer and shorter potential markers than those targeted for *Erica*. The latter may nevertheless be effective for sequence capture across the wider family Ericaceae. We tested the targets delivered by our scripts by obtaining an empirical dataset. The resulting sequence variation was lower than that of standard nuclear ribosomal markers (that in *Erica* fail to deliver a well resolved gene tree), confirming the importance of maximising the lengths of individual markers. We conclude that rather than searching for "one size fits all" universal markers, we should improve and make more accessible the tools necessary for developing "made to measure" ones.

Corresponding author
Malvina Kadlec,
mkadlec@uni-mainz.de

## INTRODUCTION

DNA sequence data is the cornerstone of comparative and evolutionary research, invaluable for inference of population-level processes and species delimitation through to higher level relationships. Sanger sequencing (*Sanger, Nicklen & Coulson, 1977*) and Polymerase Chain Reaction (PCR) amplification (*Saiki et al., 1985*) have been standard tools for decades, aided by the development of protocols that can be applied across closely and distantly related organisms. In plants, universal primers such as for plastid (*Taberlet et al., 1991*), nuclear ribosomal (*White et al., 1990*) and even single or low copy nuclear (*Blattner, 2016*) sequences have been widely applied to infer evolutionary histories. Many empirical

studies are still limited to these few independent markers, the phylogenetic signal of which may not reflect the true sequence of speciation events (*Kingman, 1982*; *White et al., 1990*). Additionally, the resulting gene trees are often poorly resolved, particularly when divergence of lineages was rapid. When it is not possible to generate a robust and unambiguous phylogenetic hypothesis using standard universal markers, protocols for alternative low copy genes are highly desirable (*Sang, 2002*; *Hughes, Eastwood & Bailey, 2006*).

With the development of next generation sequencing (NGS) techniques, we now have potential access to numerous nuclear markers allowing us to address evolutionary questions without being constrained by the generation of sequence datasets per se. In principle, the whole genome is at our disposal, but whole genome sequencing (WGS) is currently relatively expensive, time-consuming and computationally difficult, especially for non-model organisms and eukaryote genomes in general (*Jones & Good, 2016*). These disadvantages will doubtless reduce in the near future, but nevertheless much of the data that might be obtained through WGS is irrelevant for particular purposes. In the case of phylogenetic problems, repetitive elements and multiple copy genes are not useful; neither are sequences that are highly constrained and hence insufficiently variable, nor indeed those that are too variable and impossible to align; nor those subject to strong selection pressure. We need strategies to identify and target sequencing of markers appropriate for phylogenomic analysis in different clades and at different taxonomic levels, and are currently faced with an array of options.

Different methods, referred to in general as ''genome-partitioning approaches'', or ''reduced-representation genome sequencing'', have been developed that are cheaper, faster and computationally less demanding than WGS, and as such are currently more feasible for analyses of numerous samples for particular purposes (*Mamanova et al., 2010*). These include restriction-site-associated DNA sequencing (RAD-seq; *Miller et al., 2007*), and similar Genotyping by sequencing (GBS) approaches (*Elshire et al., 2011*), and whole-transcriptome shotgun sequencing (RNA-seq; *Wang, Gerstein & Snyder, 2009*). These methods can be applied to non-model species (*Johnson et al., 2012*) but do not necessarily deliver the most informative data for phylogenetic inference. RAD-seq/GBS sequences are short, generally used for obtaining (independent) single nucleotide polymorphisms (SNPs) from across the genome, suitable for population genetic analyses. RNA-seq transcriptome data cannot be obtained from dried material (such as herbarium specimens), restricting its application, and the sequences that are obtained are functionally conserved and therefore may be more suitable for analysing more ancient divergences, such as the origins of land plants (*Wickett et al., 2014*). Neither approach is ideal for inferring meaningfully resolved independent gene trees of closely related species as they will inevitably present limited numbers of linked, informative characters.

Alternative approaches can be used to target more variable, longer contiguous sequences involving selective enrichment of specific subsets of the genome before using NGS through PCR based, or sequence capture techniques. PCR based enrichment, or multiplex and microfluidic amplification of PCR products, is the simultaneous amplification of multiple targets (e.g., 48, as used in *Uribe-Convers, Settles & Tank, 2016*; to potentially hundreds
or low thousands per reaction). Although this method dispenses with the need for time-consuming library preparation, it requires prior knowledge of sequences for the design of primers; such primers must be restricted to within regions that are known to be conserved across the study group.

Current targeted sequence capture methods involve hybridization in solution between genomic DNA fragments and biotinylated RNA "baits" (also referred to as "probes" or the "Capture Library") between 70 and 120 bp long. Hybridization capture can be used with non-model organisms (as is the case for RAD-seq/GBS and RNA-seq), and shows promising results with fragmented DNA (such as might be retrieved from museum specimens) (*Lemmon & Lemmon, 2013*; *Zimmer & Wen, 2015*; *Hart et al., 2016*; *Budenhagen et al., 2016*). Moreover, even without baits specifically designed using organelle genomes, plastid and mitochondrial sequences can also be retrieved during the hybrid-enrichment process (*Tsangaras et al., 2014*). Use of targeted sequence capture for phylogenetic inference is on the increase but still somewhat in its infancy, with a range of different more or less customised laboratory and bioinformatic protocols being applied to different organismal groups and in different laboratories. The protocols follow two general approaches: One is to design baits for use in specific organismal groups (e.g., Compositae, *Mandel et al., 2014*; cichlid fish, *Ilves & Lopez-Fernandez, 2014*; and Apocynaceae, *Weitemier et al., 2014* □). To this end, conserved orthologous sequences of genes of the species of interest are identified e.g., using a BLASTn or BLASTx search (or equivalent) with transcriptome data, expressed sequences tags (ESTs) and/or WGS. Alternatively, and with considerably less effort, pre-designed sets of more universal baits are used (*Faircloth et al., 2012*; *Lemmon, Emme & Lemmon, 2012*). Of the latter, "Ultra Conserved Elements" (UCE) (*Faircloth et al., 2012*) and "Anchored Hybrid Enrichment" (AHE) (*Lemmon, Emme & Lemmon, 2012*) approaches have been applied in phylogenetic analyses of animal (e.g., snakes, *Pyron et al., 2014*; lizards, *Leaché et al., 2014*; frogs, *Peloso et al., 2016*; and spiders, *Hamilton et al., 2016*) and plant (*Medicago*, *De Sousa et al., 2014*; *Sarracenia*, *Stephens et al., 2015*; palms, *Comer et al., 2016*; *Heyduk et al., 2016*; *Heuchera*, *Folk, Mandel & Freudenstein, 2015*; *Inga*, *Nicholls et al., 2015*; and *Protea*, *Mitchell et al., 2017*) clades.

Universal protocols are an attractive prospect, in terms of reduced cost and effort, and because they might generate broadly comparable data suitable for wider analyses (or even DNA barcoding; *Blattner, 2016*). However, the resulting sequence markers may not be optimal for all purposes. For phylogenetic inference, low-copy markers are required to avoid paralogy issues, and for successful hybridisation capture similarity of baits to target sequences must fall within c. 75–100% (*Lemmon & Lemmon, 2013*). This places a restriction on more universal markers that will necessarily exclude potentially useful low copy, high variability markers where these are subject to duplications or too variable in particular lineages.

The selection of appropriate sequence markers may therefore be crucial in determining the success of this kind of analysis, especially for non-model species. Transcriptome data for increasing numbers of non-model organisms are available (*Matasci et al., 2014*) and can already be used for marker selection in many plant clades. Bioinformatics tools are available that can assist in the selection of markers and design of baits, taking transcriptome and/or

whole genome sequences of relevant taxa as input. These include MarkerMiner (*Chamala et al., 2015*), Hyb-Seq (*Weitemier et al., 2014*; *Schmickl et al., 2016*) and BaitsFisher (*Mayer et al., 2016*). The question for researchers embarking on phylogenomic analyses is whether it is worth the additional cost and effort involved in designing custom baits, and how to select sequence markers in order to get the most information out of a given investment of time and funds.

Our ongoing research addresses the challenge of resolving potentially complex phylogenetic relationships between closely related populations and species of a non-model flowering plant group, the genus *Erica* (Ericaceae; one of 22 families of the asterid order Ericales; *Stevens, 2001*). The c. 700 South African species of *Erica* represent the most species rich 'Cape clade' in the spectacularly diverse Cape Floristic Region (*Linder, 2003*; *Pirie et al., 2016*). Analyses of the *Erica* clade as a whole offer a rich source of data in terms of numbers of evolutionary events, and our ability to infer such events accurately is arguably greatest in the most recently diverged species and populations. In such clades, the historical signal for shifts in key characteristics and geographic ranges are in general less likely to have been overwritten by subsequent shifts and (local) extinction. However, phylogenetic inference in rapid species radiations, such as that of Cape *Erica* (*Pirie et al., 2016*), Andean *Lupinus* (*Hughes & Eastwood, 2006*) or Lake Malawi cichlid fish (*Santos & Salzburger, 2012*) presents particular challenges. These include low sequence divergence confounded by the impact of both reticulation and coalescence on population-level processes. To infer a meaningful species tree under such circumstances, we need data suitable to infer multiple, maximally informative, independent gene trees.

The aims of this paper are to compare the performance of custom versus more universal approaches to marker selection for groups of closely related species/populations. Applying new scripts and a number of similar existing methods for marker selection, we compare predicted sequence lengths and variability of the resulting markers as estimates of their potential for delivering multiple independent and informative gene trees. We further compare different options implemented in our scripts for optimising e.g., intron numbers/lengths for a given number of baits. In so doing, we generate a tool for low-level phylogenetic inference in *Erica*, we test it experimentally by generating empirical data, and we assess its potential application across a wider group, e.g., the family Ericaceae.

## MATERIALS & METHODS

Our first aim was to identify homologous, single-copy sequence markers for which we could design baits (probes) with similarity of ≥75% (as hybridization between target and probe tolerates a maximum of 25% divergence) that would be predicted to deliver the greatest numbers of informative characters. Baits currently represent a relatively large proportion of the total cost of the protocol (which is expensive on a per sample basis compared to e.g., PCR enrichment). We therefore restricted the total length of hybridisation baits to 692,400 bp (5,770 individual 120 bp baits), representing a total "capture footprint" (i.e., cumulative sequence length) of 173,100 bp given probe overlap representing 4x coverage. With our lab protocol (see below) this permits dilution of the baits to capture

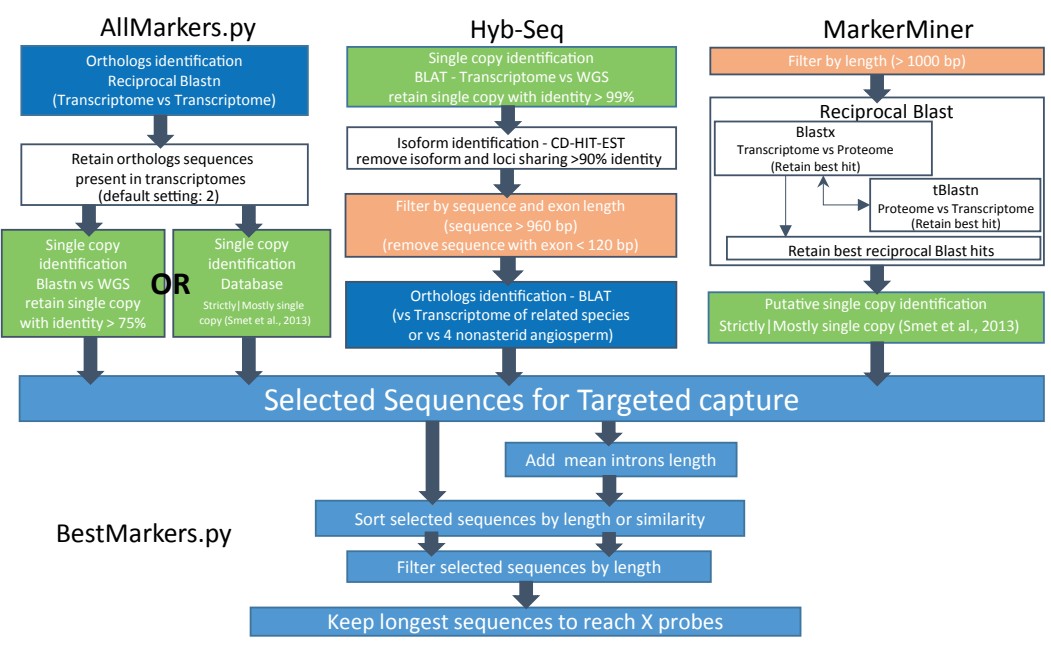

**Figure 1** Flowchart(s) illustrating the methods used for marker selection.

five samples per unit of baits instead of just one. We developed custom-made Python 2.7.6 scripts to identify the wider pool of all potential target sequences from transcriptome and WGS data (both of which were available from published sources; details below), as well as applying already available scripts/software for comparison. We subsequently implemented in further scripts different options for prioritising target variability, length and/or intron numbers and lengths to select optimal sequence markers from these pools of potential targets. We then compared the lengths and numbers of the sequences in the different resulting potential and optimal marker sets.

## Identifying potential target sequences

Our custom-made script (AllMarkers.py; summarised in Fig. 1 available at Github: https://github.com/MaKadlec/Select-Markers/tree/AllMarkers) requires at least two transcriptomes, ideally of taxa closely related to the focal group. Where WGS/genome skimming data of one or more such taxa is available, it can be used too, as in *Folk, Mandel & Freudenstein (2015)*. AllMarkers.py implements the following steps: First, two or more transcriptomes are compared to identify homologues, retaining those found in at least two transcriptomes. These are hence likely to also be found in related genomes. We have successfully used up to eight transcriptomes; on eight cores of a fast desktop PC the analyses ran for up to two days. Particularly when larger numbers of larger transcriptomes are compared, an additional filter can be applied prior to this step to remove shorter sequences (e.g., those <1,000 bp) and thereby improve speed. Next, multiple copy sequences are identified, for which homology assessment might be problematic. When WGS data is available, this is achieved using BLASTn of transcriptome against WGS. When no WGS data is available it is by comparison to the classification of proteins as single/mostly single

copy across angiosperms by *De Smet et al. (2013)*, using BLASTx following the approach used in MarkerMiner (*Chamala et al., 2015*). Multiple-copy sequences are then excluded. Finally, a filter for similarity ≥75% is applied. This series of steps is comparable to but differs from those implemented in Hyb-Seq (*Weitemier et al., 2014*) and in MarkerMiner (*Chamala et al., 2015*) (Fig. 1), which we also applied here.

The Hyb-Seq pipeline uses transcriptome and WGS sequences of closely related species to select marker sequences. This pipeline employs BLAT (BLAST-like Alignment Tool), rather than BLAST as in AllMarkers.py, to identify single-copy sequences with identity >99%. After isoform identification, sequences with exons <120 bp and those of total length <960 bp are removed. This represents a further filtering of potential targets that is comparable in part to the next steps in our own scripts, as described below. Then orthologous sequences are identified using a transcriptome of a closest related species or of one of four angiosperms (*Arabidopsis thaliana*, *Oryza sativa*, *Populus trichocarpa* and *Vitis vinifera*), as opposed to by comparison to two or more transcriptomes in AllMarkers.py.

For MarkerMiner, WGS data is neither required (as in HybSeq) nor used if available (as in AllMarkers.py). This pipeline involves selecting sequences by size in input transcriptomes (we set length parameter to >1,000 bp) then using reciprocal BLAST between transcriptomes and a reference proteome to select sequences above 70% similarity. The proteome most closely related to *Erica* implemented in MarkerMiner in August 2016 was that of *Vitis vinifera* (Vitaceae; Vitales; core eudicots; *Stevens, 2001*). This minimum similarity threshold does not directly reflect that required for successful probe hybridisation, and particularly given comparison to a relatively distantly related proteome (as opposed to more closely related transcriptomes with AllMarkers.py and HybSeq) can be expected to be conservative. In the final step, MarkerMiner retains putative single copy ortholog pairs following *De Smet et al. (2013)*, as also implemented in AllMarkers.py when no WGS is available.

## Selection of optimal target sequences from pools of potential targets

The above steps result in potentially large pools of potentially highly suboptimal targets, in particular shorter and/or invariable sequences that, given rapid lineage divergence, may not deliver enough informative characters to discern meaningfully resolved independent gene trees. In order to select optimal markers from these pools given a limited number of baits we designed a further script (available at Github: https://github.com/MaKadlec/Select-Markers/tree/BestMarkers.py). Depending on the phylogenetic problem to hand (e.g., recent, species level divergence versus older radiations) and available information (e.g., about sequence variability in the focal clade; positions and lengths of potentially more variable introns), various options are possible. In our case, from WGS and transcriptome data we know where introns are likely to be found, but in the absence of sequences from multiple accessions of our ingroup, the only indication of sequence variability comes from comparison of coding regions of relatively distantly related taxa, i.e., single species of *Rhododendron*, *Vaccinium* and *Erica*. We therefore assessed two options: (1) simply selecting the longest sequences. (2) Selecting the longest sequences, but taking into account the (likely) additional length of introns. Using WGS data, we assessed

the number and length of introns. For the purpose of ranking potential markers, we decided to use mean intron length in order to avoid favouring the selection of sequences with large introns that (a) might not be efficiently captured/sequenced; or (b) might not be so large in the focal clade. Finally, the longest sequences were selected that could be captured with our maximum number of baits. Coding regions <120 bp long are shorter than the baits and are likely to be ineffectively captured. For this reason, in the Hyb-Seq approach (*Weitemier et al., 2014*) all sequences including exons <120 bp are excluded; however, this is at the expense of excluding otherwise optimal markers that may include individual exons of <120 bp. We therefore opted to retain sequences including one or more coding regions ≥120 bp, whilst excluding individual exons <120 bp as potential targets for baits.

## In silico comparison with empirical data

Our custom scripts (AllMarkers.py and BestMarkers.py), the Hyb-Seq and MarkerMiner pipelines were each applied to transcriptomes and (except for MarkerMiner) WGS of representatives of the Ericaceae subfamily Ericoideae. Transcriptome data was of *Rhododendron scopulorum* (18,307 gene sequences; 1KP project; *Matasci et al., 2014*) and (diploid) cranberry *Vaccinium macrocarpon* (48,270 sequences; PRJNA260125). WGS was of *V. macrocarpon* (PRJNA246586) and *Erica plukenetii* (Le Maitre & Bellstedt, preliminary data; sequences matching the selected markers included in Data S1). We compared the (potential) length and identity of the resulting targets.

We then compared these "made to measure" (*Erica*/Ericoideae-specific) targets with those that might be selected using a more "one size fits all" (universal) approach to probe design. For this purpose, we used transcriptomes from increasingly distantly related plants as available on NCBI. First we included different families of the wider order Ericales: Actinidiaceae (*Actinidia chinensis*; 10,000 sequences; PRJNA277383), Primulaceae (*Aegiceras corniculatum*; 49,412 sequences; PRJNA269022), Theaceae (*Camellia reticulata*; 139,145 sequences; PRJNA297756), Ebenaceae (*Diospyros lotus*; 413, 775 sequences; PRJNA261339), and Ericaceae (*R. scorpulum and V. macrocarpon*, as above). Then we expanded to different orders of eudicots: Ranunculales (*Anemone flaccida*; 46,945 sequences; PRJNA277332), Asterales (*Dahlia pinnata*; 35,638 sequences; PRJNA189243), Proteales (*Gevuina avellana*; 185,089 sequences; PRJNA299715), Caryophyllales (*Mesembryanthemum crystallinum*; 24,204 sequences; PRJNA217685), Solanales (*Solanum chacoense*; 42,873 sequences; PRJNA299204), Fabales (*Vigna radiata*; 78,617 sequences; PRJNA266360), Vitales (*Vitis vinifera*; 52,310 sequences; PRJNA239278) and Ericales (*R. scorpulum*, as above). Because in this wider context it is no longer appropriate to identify single copy markers on the basis of Ericoideae data alone, we instead used the option to compare to the angiosperm-wide database (*Smet et al., 2013*) following an approach similar to MarkerMiner (*Chamala et al., 2015*). We compared the resulting targets to those of the *Erica*-specific approach, as above.

## Generation of a novel empirical dataset

In order to confirm that our scripts can be used to obtain datasets of single-copy markers, we applied them to our empirical study on Cape *Erica*. We used the 132 sequences

**Table 1 Voucher details.** Samples used for DNA extraction and their collection localities. Vouchers were lodged at herbarium NBG (MP: Pirie).

| Voucher | Sample # | Species | Locality (unless specified, within the Western Cape, South Africa) |
|---|---|---|---|
| MP1320 | 78 | E. abietina L. ssp. aurantiaca | Du Toit's Pass |
| MP1330 | 74 | E. coccinea L. | RZE, Greyton |
| MP1336 | 81 | E. coccinea L. | Groot Hagelkraal |
| MP1318 | 72 | E. imbricata L. | Flouhoogte |
| MP1319 | 73 | E. imbricata L. | Stellenbosch |
| MP1334 | 74 | E. imbricata L. | Groot Hagelkraal |
| MP1311 | 69 | E. imbricata L. | Boskloof |
| MP1312 | 80 | E. lasciva Salisb. | Boskloof |
| MP1325 | 83 | E. lasciva Salisb. | Albertinia |
| MP1309 | 71 | E. penicilliformis Salisb. | Boskloof |
| MP1339 | 75 | E. placentiflora Salisb. | Cape Hangklip |
| MP1333 | 82 | E. plukenetii L. | Groot Hagelkraal |
| | 68 | R. camtschaticum Pall. | Oldenburg Botanical Garden, Germany (cultivated) |

resulting from our custom scripts, taking into account the potential intron lengths (see Results and Discussion).

In addition to these targets, we added two additional markers that were not otherwise selected as optimal, for the purpose of comparison with other datasets. These were rpb2 (as used in phylogenetic reconstruction in *Rhododendron; Goetsch, Eckert & Hall, 2005*) and topoisomerase B (as proposed for use across flowering plants; *Blattner, 2016*).

*Laboratory methods*: Plant material was collected in the field under permit (Cape Nature: 0028-AAA008-00134; South Africa National Parks: CRC-2009/007-2014) or obtained from cultivation. DNA was extracted from one sample of *Rhododendron camtschaticum*, supplied by Dirk Albach and Bernhard von Hagen from collections of the Botanic Garden, Carl von Ossietzky Universität, Oldenburg, Germany; and 12 of *Erica* (Table 1) using Qiagen DNAeasy kits (Qiagen, Hilden, Germany). DNA extraction in *Erica* is generally challenging (*Bellstedt et al., 2010*) and the quantity and quality of DNA obtained differed considerably between species. To reach the correct amount of DNA required for library preparation, multiple DNA extractions from the same sample were combined.

For library preparation and hybridisation enrichment, we used the Agilent SureSelectXT protocol (G7530-90000), incorporating sample-specific indexes for pooled sequencing, with a 1kb-499kb SureSelectXT *Custom* capture library designed using the SureDesign Custom Design Tool for NGS Target Enrichment, specifying 4× coverage and probe length 120 bp. For the library preparation, amount of gDNA used was between 1 and 3 μg, and during the hybridisation and capture step, we used a diluted capture library (1 part Agilent baits solution to 4 of ddH$_2$O). Sequencing was performed with Illumina NextSeq500 (StarSeq, Mainz, Germany) to generate 25 million paired-end reads of length 150 bp.

### Bioinformatic analysis

As the total footprint of the capture library (the cumulative sequence lengths of all the selected markers) was small, *de novo* assembly was possible. We chose to use MIRA

(version 4.0) (*Chevreux, Wetter & Suhai, 1999*), in part because MIRA can be used to perform both *de novo* assembly and mapping. The two options were used with default parameters for Illumina (overlap value = 80 for *de novo* and 160 for mapping assembly; quality level = accurate). Reads were assembled into contiguous sequences (contigs). We then compared using BLASTn against the sequence targets (complete sequences and coding region sequences) as well as against nuclear ribosomal (nrDNA), plastid, and mitochondrial data. Contigs for which overlap with targets was under 100 bp and similarity to target sequences was less than 75% were removed. Using the L-INS-i (iterative refinement method incorporating local pairwise alignment information) method of MAFFT (*Katoh et al., 2002*), we aligned contigs with each other and with the sequence targets (complete sequences and coding region sequences). Contigs were checked with Gap5 (*Bonfield & Whitwham, 2010*) and by comparison to the alignments to identify and confirm remaining separate overlapping contigs without sequence differences. We used custom made scripts to merge and remove redundant contigs, combining only those with identical overlapping sequences (minimum overlap of 30 bp) or which differed by a single base only (in which case this position was coded with IUPAC ambiguity codes). Contigs differing by more than one base, or which did not overlap, were not combined. This should avoid combining non-continuous contigs representing different copies or alleles, at the cost of tending to overestimate the numbers of such copies where overlap of contigs is incomplete. We then attempted to add to the alignments any <100 bp sequences or sequences under 75% similarity that matched the target according to BLASTn, combining (or not) contigs using the same principles as above.

We excluded alignment positions representing indels or missing data in one or more samples and then calculated the percentage of variable sites per marker, including combined mitochondrial and plastid sequences and individual nrDNA sequences representing Internal and External Transcribed Spacer regions (ITS and ETS) as obtained using Sanger sequencing in previous work (*Pirie, Oliver & Bellstedt, 2011*; *Pirie et al., in press*). Gene trees were inferred using RAxML (*Stamatakis, 2014*) and used as a rough test for potential paralogy, under the assumption that the ingroup (comprising all samples except *Rhododendron* and the more closely related outgroups *Erica abietina* and *Erica plukenetii*) is monophyletic. We summarised 70% bootstrap consensus trees using DendroPy (*Sukumaran & Holder, 2010*) with SumTrees (https://github.com/jeetsukumaran/DendroPy).

## RESULTS

### Similarity, length and overlap of selected markers: "made to measure" versus "one size fits all"

The lengths of sequences selected using the different scripts are presented in Fig. 2. Summary comparisons by method are presented in Table 2 (sequence numbers, lengths and similarity). In general, the additional filter that includes mean intron length resulted in an increased number of shorter targets that might nevertheless deliver greater final sequence lengths, if average lengths of flanking introns are effectively captured (Fig. 2).
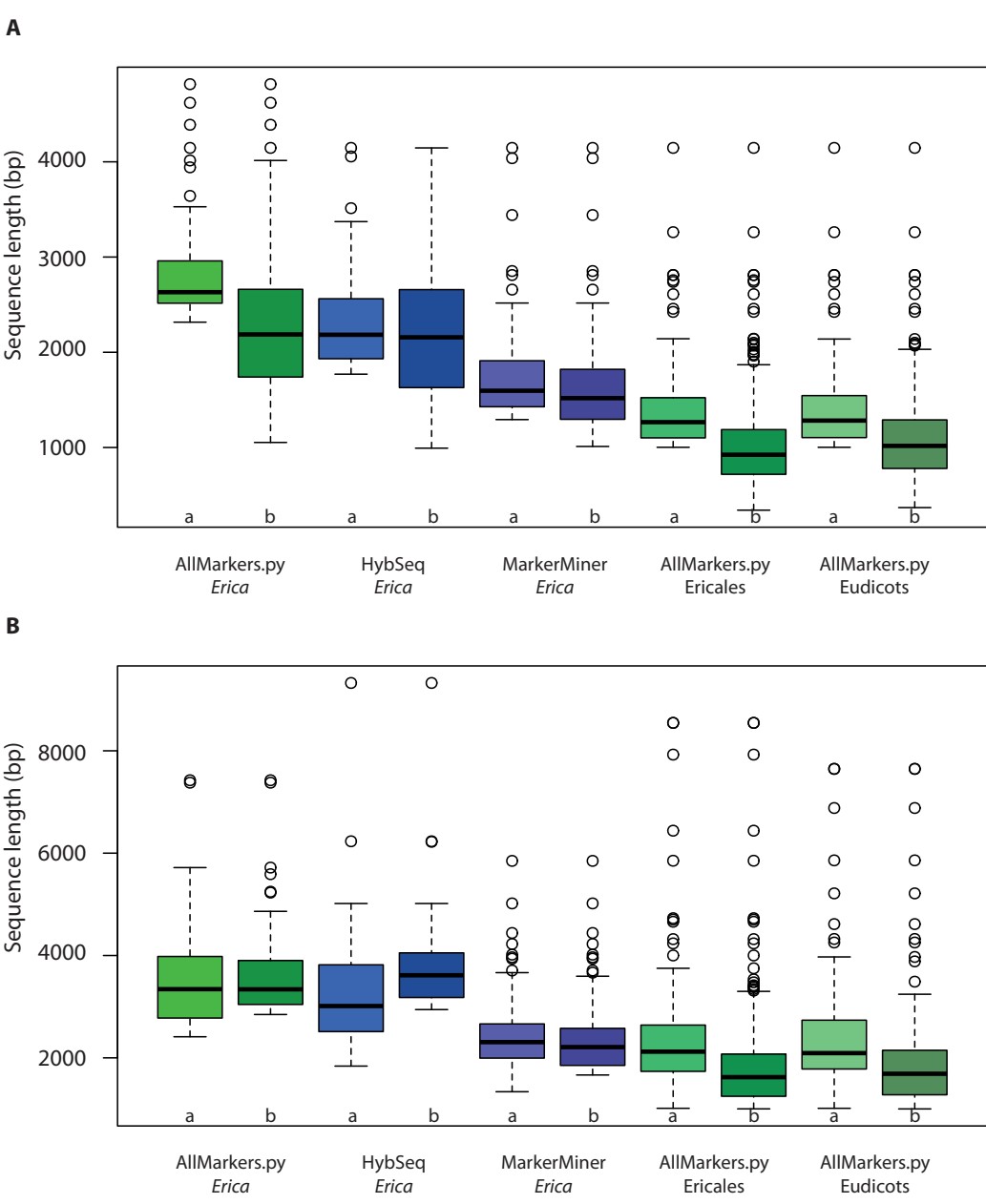

**Figure 2** **Selected exon/predicted marker lengths by method.** Summary of (A) exon lengths and (B) predicted exon plus intron lengths of markers selected using AllMarkers.py (shades of green), Hyb-Seq (blue) and MarkerMiner (purple) followed by BestMarkers.py. Each pair of plots represents the markers selected when optimising for exon lengths (a) and predicted exon plus intron lengths (b).

## Made to measure

We identified 4,649 potential markers using our custom script AllMarkers.py. Applying script BestMarkers.py to this pool to optimise for length, two different subsets of optimal markers were obtained: 132 with median length (of coding region) of 2,187 bp when taking intron lengths into account; 79 of median length 2,631 bp when not. Sequence identity was similar (Table 2).

**Table 2 Attributes of selected markers.** Range, median and average length of selected markers in *Rhododendron*, with and without taking introns into account, and similarities to homologues in *Vaccinium*.

| | | Length of CR (bp) | | Similarity (%) | | Predicted length (bp) | |
|---|---|---|---|---|---|---|---|
| | | Range | Mean Median sd | Range | Mean Median sd | Range | Mean Median sd |
| | AllMarkers.py (without intron length)—79 seq | 2,316–4,815 | 2,834 2,631 535 | 82–96 | 90 90 3.1 | 2,412–7,425 | 3,541 3,342 998 |
| | AllMarkers.py (with intron length)—132 seq | 1,053–4,815 | 2,287 2,187 736 | 81–97 | 91 92 3.5 | 2,847–7,425 | 3,579 3,339 773 |
| *Erica* | HybSeq (without intron length)—66 seq | 1,170–4,146 | 2,350 2,184 549 | 77–95 | 89 91 5 | 1,839–9,326 | 3,285 3,013 1,181 |
| | HybSeq (with intron length)—55 seq | 993–4,146 | 2,226 2,157 719 | 77–95 | 89 91 5 | 2,943–9,326 | 3835 3,614 1,032 |
| | MarkerMiner (without intron length)—207 seq | 1,293–4,146 | 1,726 1,596 419 | 85–97 | 93 94 2 | 1,338–5,849 | 2,411 2,307 649 |
| | MarkerMiner (with intron length)—254 seq | 1,011–4,146 | 1,600 1,518 454 | 85–97 | 93 94 2 | 1,665–5,849 | 2,329 2,210 611 |
| Ericales | AllMarkers.py (without intron length)—171 seq | 1,002–4,146 | 1,400 1,266 460 | 82–97 | 93 93 2.6 | 1,014–8,546 | 2,389 2,121 1,153 |
| | AllMarkers.py (with intron length)—408 seq | 342–4,146 | 1014 924 458 | 82–97 | 93 93 2.3 | 1,003–8,546 | 1830 1,623 928 |
| Eudicots | AllMarkers.py (without introns length)—130 seq | 1,002–4,146 | 1,427 1,283 487 | 85–97 | 93 93 2.4 | 1,014–7,657 | 2,379 2,093 1,089 |
| | AllMarkers.py (with introns length)—249 seq | 369–4,146 | 1,112 1,017 494 | 85–97 | 93 94 2.2 | 1,002–7,647 | 1,895 1,689 960 |

With the Hyb-Seq pipeline, 782 sequences were obtained, which after applying BestMarkers.py, was reduced to 55 of median length 2,157 bp when taking introns into account and 66 of median length 2,184 bp when not. Sequence identity was similar, and similar to that resulting from AllMarkers.py (Table 2).

With MarkerMiner, target sequences are delivered separately for each transcriptome provided. We selected a total pool of 544 potential target sequences, of which 389 are represented in the *R. scopulorum* data and 222 in *V. macrocarpon*. We identified just 67 that were common to both (whereby it should be noted that AllMarkers.py by default retains

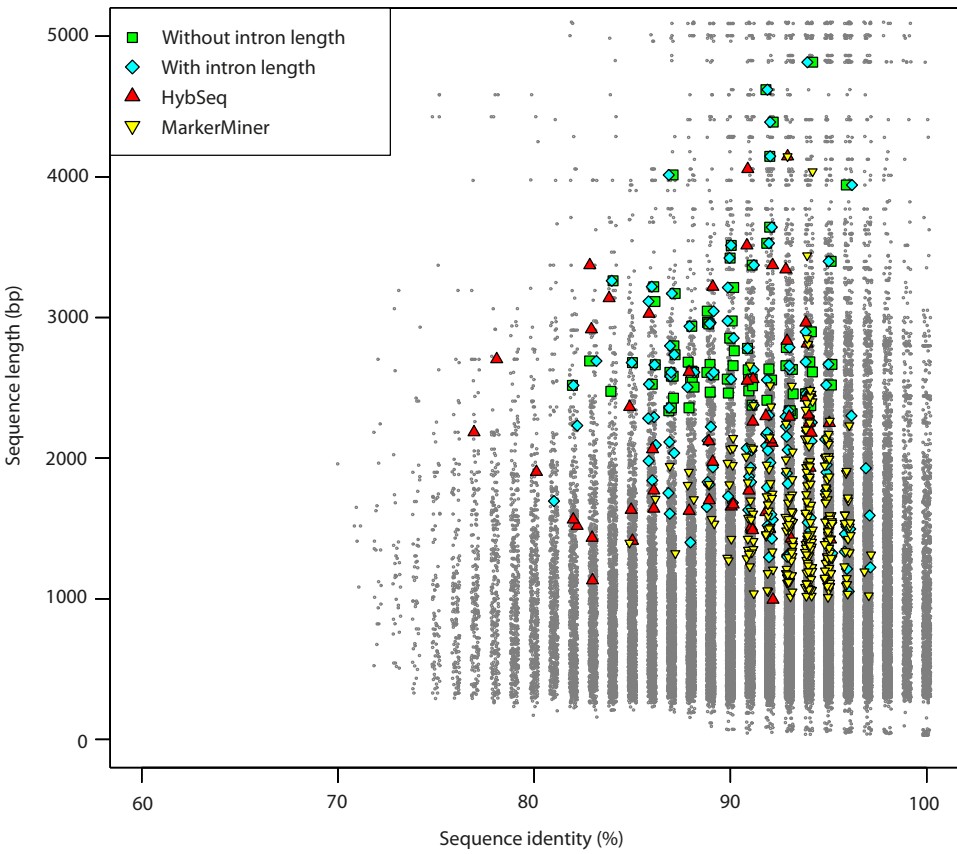

**Figure 3** Length versus variability of potential sequence markers (grey dots) and those selected using BestMarkers.py from the pools generated by the different methods (coloured symbols).

only those found in at least two transcriptomes). Of the 544 sequences, 519 are indicated by MarkerMiner as mostly single copy and 25 as strictly single copy in angiosperms. After applying BestMarkers.py we retained 254 sequence targets when taking introns into account and 207 sequences when not. Use of MarkerMiner resulted in the selection of greater numbers of shorter and slightly more conserved markers compared to both AllMarkers.py and HybSeq (Table 2, Figs. 2 and 3).

### One size fits all

Applying AllMarkers.py/BestMarkers.py to transcriptomes of Ericales resulted in a pool of 2,354 potential markers and final datasets of 409 sequences when taking introns into account and 171 when not. With the Eudicot transcriptomes, the total pool included 461 potential markers and final datasets 249 (when taking introns into account) and 130 sequences (when not) (Table 2). In the latter, there is a slight increase in similarity ($\geq 85\%$, similar to MarkerMiner; Fig. 3), and in both, sequences are shorter (Table 2 and Fig. 2).

The numbers of markers in common given the different methods for selecting them, before and after applying BestMarkers.py are presented in Fig. 4. Figure 4A illustrates both the low overlap and large differences in numbers between the complete pools of potential markers identified using the different methods/input data. Expanding in taxonomic

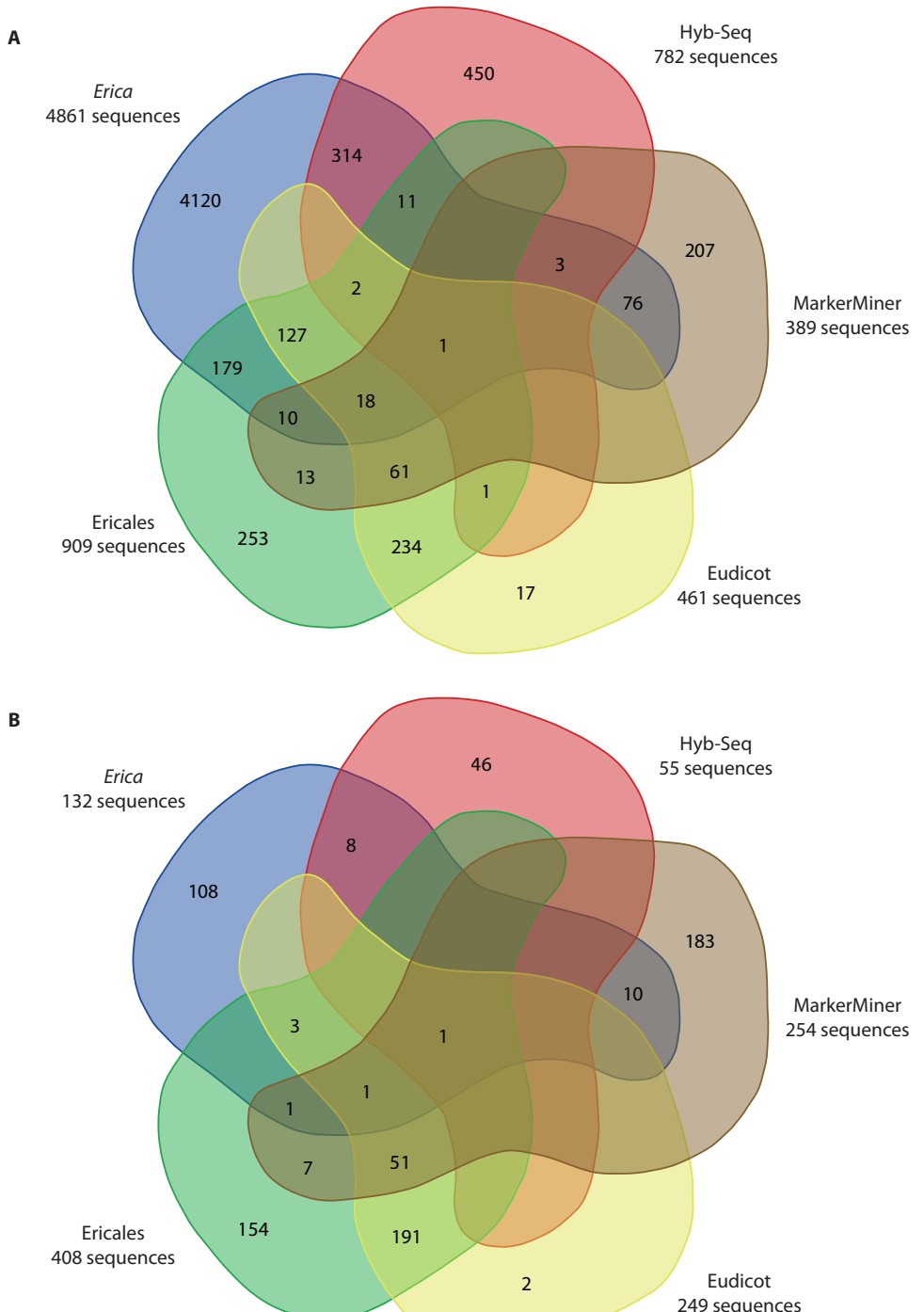

**Figure 4 Overlap of selected markers by method.** Venn diagrams produced using http://bioinformatics. psb.ugent.be/webtools/Venn/comparing overlap in markers selected given the different methods, superimposed with their numbers. (A) The complete pools of potential markers; (B) the subsets of markers selected using BestMarkers.py, optimising for total predicted length (exons and introns).

scope from *Erica* (identifying single-copy genes on the basis of WGS data) to Ericales and to eudicots (adopting single copy markers from the database of *De Smet et al. (2013)* resulted in a decrease in numbers of potential markers, and the use of MarkerMiner a further decrease. Figure 4B illustrates the differences in the optimal markers selected using BestMarkers.py on these pools. There is limited overlap and considerable differences in both target numbers and lengths: overall, AllMarkers.py/BestMarkers.py and HybSeq delivered the longest sequences, whereby the former delivered more markers for the same number of baits. Both the Ericales and eudicot analyses and MarkerMiner delivered greater numbers of shorter sequences.

## Empirical data

We performed selective enrichment of 134 markers (132 selected using AllMark-ers.py/BestMarkers.py, plus the two 'universal' markers added for the purposes of comparison). Exon sequences used for probe design are presented in Data S1 and sequence alignments in Data S2. Raw sequence reads are deposited on NCBI (PRJNA388814). With the exception of a single marker, capture was equally effective in the single *Rhododendron* sample and thirteen *Erica* samples. One marker was captured only in *Rhododendron*, and two others was not captured at all. All of the remaining 129 markers plus rpb2 and topoisomerase B were recovered, at least in part, from all thirteen samples analysed (Data S3). Of these, six were single copy without allelic variation; 83 included sequence polymorphisms corresponding to two distinguishable putative alleles in one or more (but not all) individual samples. A further 40 included sequence polymorphisms in all samples, which exhibited two or more copies. Of the latter 40, 28 represented paralogs that were easily distinguished on the basis of high sequence divergence in one or more coding region(s) and could thus be segregated into separate matrices of homologous sequences. The remainder (12) included multiple contigs that could not obviously be combined into single homologous sequences or pairs of alleles. Inspection of individual gene trees (Data S4) failed to reject the monophyly of the ingroup in all but five cases.

Comparison of sequence length/variability was limited by uneven sequencing coverage, but we could confirm the capture of complete intron sequences of up to c. 1,000 bp and partial introns/flanking non-coding regions of up to c. 500 bp. In addition, large stretches of homologous high copy nuclear ribosomal and mitochondrial sequences were captured for all samples, as well as more fragmented plastid sequences.

Despite incomplete sequencing coverage, the average alignment length of single copy nuclear sequences was 1,810 bp, with a range between 823 and 5,574 bp. With all gaps and missing data excluded (resulting in alignments of between 327 and 4,716 bp), the single copy nuclear sequences in the ingroup presented between five and 412 variable positions each, representing a range of 2.6–26.1% variability. Variability of rpb2 was 3.4%; topoisomerase B: 7.5%; ETS: 22.1%; ITS: 17.9%; mitochondrial: 6.3%; and plastid sequences: 0.54%. A plot of original predicted length of markers (instead of real length since in most cases complete sequences were not obtained) against variability is presented in Fig. 5. There was no obvious relationship between sequence length and variability. A further plot of observed sequence variability against variability of the corresponding transcriptome data

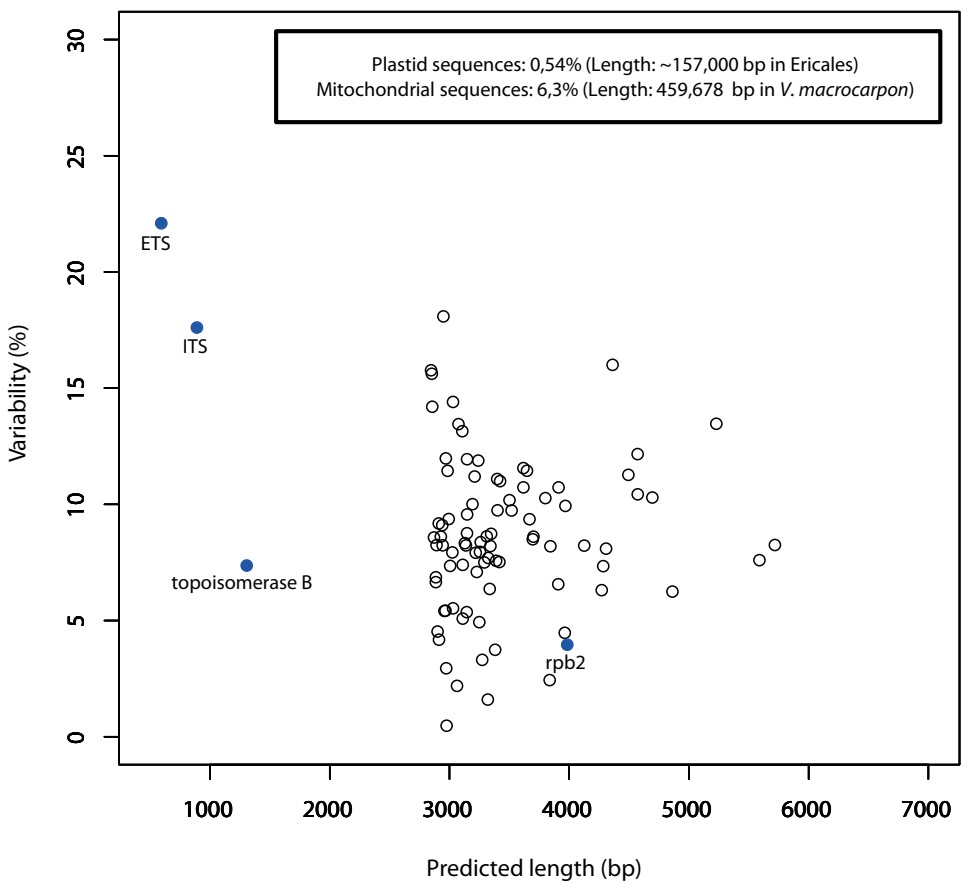

**Figure 5** **Sequence variability observed in the empirical data plotted against predicted sequence length.** "Universal" markers rpb2 and topoisomerase B are indicated and plastid, mitochondrial and nrDNA are included with indication of sequence lengths derived from the literature.

(*Rhododendron* compared to *Empetrum*) is presented in Data S5; there was also no obvious relationship. Gene trees inferred under ML are documented in Data S3 (with further details in Data S4), with eight based on selected markers (ITS, mitochondrion, and six single copy nuclear markers that delivered the greatest numbers of clades supported by ≥70% BS) illustrated in Data S6.

## DISCUSSION

### Comparing closely versus distantly related genomes for marker selection

It seems intuitively obvious that optimal markers for a given phylogenetic problem will be those informed by comparison to transcriptomes/WGS of the most closely related representative taxa. With such data, lineage specific gene duplications can be identified and the number of potential targets of appropriate variability maximised. However, the genomic data available for a given focal group (such as transcriptome data from the 1KP project; (*Matasci et al., 2014*) may represent taxa more or less distantly related to it, and particular researchers may or may not wish to go to the trouble of designing and

applying custom protocols. Indeed, if an off-the-shelf tool will provide appropriate data, it would be a great deal simpler just to use it. Hence, before embarking on expensive and time-consuming lab procedures, we need to know to what degree targets designed for one group might be applied to more distantly related ones (e.g., in this case the utility of *Erica* baits across Ericaceae, or Ericales); and conversely, how suboptimal baits designed for universal application (e.g., across angiosperms) are likely to be for a given subclade.

Using our own custom scripts, we compared the pools of markers that might be selected on the basis of comparison of relatively closely related genomes with those on the basis of more distantly related ones (i.e., within the subfamily Ericoideae as opposed to within the order Ericales or across eudicots). Our results showed that both the pools and the best marker sets from those pools differed considerably, and that the sequences of the latter were considerably shorter (Table 2, Figs. 2 and 3). On the other hand, sequence variability within Ericales (minimum sequence identity between Ericaceae and Actinidiaceae: 73%) suggests that baits designed for *Erica* are also potentially suited for use at least across Ericaceae, including in *Rhododendron* and *Vaccinium* (both species-rich genera for which such tools might be particularly useful (*Kron, Powell & Luteyn, 2002*; *Goetsch, Eckert & Hall, 2005*). In general, our results confirm both the greater potential of custom baits developed for specific clades; and show that once obtained, such tools are nevertheless likely to apply across a fairly broad range of related taxa.

### The impact of method for marker selection

Having decided to design custom baits, the next question that we might ask is which method to use for probe selection/design. Our results suggest that this is also likely to have a significant impact on the resulting datasets. We compared three approaches to marker selection: our own custom scripts; those presented in the Hyb-Seq approach (*Weitemier et al., 2014*) and MarkerMiner (*Chamala et al., 2015*).

Of these three, MarkerMiner is arguably the most user-friendly, which is important given that its user base ought ideally to include biologists without extensive bioinformatics skills. However, in our comparisons it delivered the shortest sequence lengths (Table 2). The reasons for this are two fold. First (and perhaps most importantly), because the transcriptomes used, irrespective of their similarity one to another, are compared to what is likely to be a rather distantly related proteome. Second, because the approach for identifying single or low-copy markers involves comparison to a general database (in this case for flowering plants), rather than a case-by-case assessment. Hence, in the current implementation of MarkerMiner it is to be expected that the most variable sequences will be excluded. So will some that are single copy in the focal group (or with easily discerned paralogs, as was the case here and also at lower taxonomic levels in *Budenhagen et al., 2016*), but not in other clades; and some that are multiple-copy may in fact be included. This is reflected in our results by the low number of potential target sequences recovered in total; in the low proportion of those that were recovered also being recovered using our own custom scripts and Hyb-Seq; and in the lower sequence length: the removal of more variable sequences arbitrarily results in the removal of longer ones too (Table 2). This phenomenon is apparently also reflected in the even shorter sequences reported by

*Budenhagen et al. (2016)*, using universal angiosperm probes (average 764 bp, derived from targets averaging 343 bp).

The Hyb-Seq approach is more similar to our own, but nevertheless results in a different dataset of selected sequences. The main differences lie in the search tool and filters. Our script uses BLAST, whereas Hyb-Seq uses BLAT. BLAT is faster than BLAST, but needs an exact or nearly-exact match to return a hit. Significantly, the exclusion in HybSeq of all sequences including any exons <120 bp is at the loss of markers including variable introns; in our approach the problem of short exon/probe mismatch is avoided simply by ignoring such exons during probe design. The net result is that while both approaches deliver long target sequences, ours can deliver those including more introns (which can therefore be captured using fewer baits).

## Selecting optimal markers from within a pool of potential candidates

Our approach includes not just a means to select potentially appropriate markers (AllMarkers.py; as is the case with the other approaches compared) but also a second step (BestMarkers.py) that selects putatively optimal markers from amongst that pool. Obviously, it is possible to capture and sequence the entire pool (following *Ilves & Lopez-Fernandez, 2014*; *Mandel et al., 2014*; *Weitemier et al., 2014*). However, by targeting a smaller number of the most appropriate markers, more samples can be analysed less expensively. A given bait solution can be used for a greater number of samples (because it includes fewer different baits, each at higher concentration), whilst sequencing effort can be reduced by eliminating a potentially large number of less informative (or perhaps even entirely uninformative) markers.

AllMarkers.py identifies and reports the positions of introns from comparison of WGS to transcriptome data. Subsequently optimising for intron numbers/length, as implemented in BestMarkers.py, would seem appropriate for the purpose of identifying regions that are likely to be both longer and more variable (*Folk, Mandel & Freudenstein, 2015*). Hybrid capture can result in sequencing of potentially long stretches of flanking regions (*Tsangaras et al., 2014*) without requiring matching baits, and introns should be less constrained, possibly with informative length variation too. Hence, taking into account the additional length of introns in marker selection can result in greater numbers of longer (and likely more variable) obtained sequences. Our empirical results support this approach: sequences showed intron capture of up to 1,000 bp, including regions in which multiple introns are interspersed with short (<120 bp) exons for which no probes were used. Intron sequences from WGS data can nevertheless be included in the output of AllMarkers.py and used to design probes. This may be effective at low taxonomic levels when WGS appropriate to assess sequence similarity within the focal group is available. Alternatively, if the problem to be addressed represents older divergences (e.g., phylogenetic uncertainty within Ericaceae; (*Freudenstein, Broe & Feldenkris, 2016*) for which length variation in introns would be unhelpful, BestMarkers.py can be used to optimise the length of exons alone.

An alternative to optimising for sequence length (with or without taking introns into account) would be to optimise for variability (or combined length and variability). We included this option in BestMarkers.py, but in the absence of data with which to

compare within our ingroup, decided *a priori* that we would be more likely to optimise total per sequence variation by selecting on the basis of length alone. This decision was supported by the empirical results: as might be expected, there was no obvious relationship between sequence length and variability (Fig. 5) and the numbers of informative characters provided by a given target could not be predicted from the similarity of the *Vaccinium* and *Rhododendron* transcriptomes (Data S5).

The variability of the data we obtained can be compared to that of nrDNA, plastid and mitochondrial sequences (and which were also obtained here without the need for matching baits due to their high copy number) and to two generally single copy nuclear genes, topoisomerase B and rpb2 (Fig. 5). Consistent with the results presented by *Nicholls et al. (2015)*, the variability of the nrDNA spacer regions (ITS and ETS) that are frequently used in empirical studies of plants is at the upper end of that observed in the sequences we obtained (of which topoisomerase B and rpb2 were fairly typical); plastid (and mitochondrial) sequences at the lower end. Given the comparably modest variability of most alternative nuclear markers, this suggests that even in cases where ITS/ETS present sufficient information to infer a well resolved nrDNA gene tree (not the case in Cape *Erica*, *Pirie, Oliver & Bellstedt, 2011*; Data S6), considerably longer sequences will be needed to infer comparably resolved independent gene trees. Difficult phylogenetic problems arise when gene trees can be expected to differ, but those inferred from standard markers are not sufficiently resolved to actually reveal it. Low information content of individual markers limits accuracy of species tree inference methods (*Lanier, Huang & Knowles, 2014*), and when relationships are contentious, resolution can be influenced disproportionately by small numbers of individual markers or sites (*Shen, Hittinger & Rokas, 2017*). These are the cases for which targeted capture approaches offer the greatest potential. We need to target markers that might deliver a forest of trees, rather than just more bushes, and not all targeted enrichment strategies are optimised to deliver this kind of data.

## CONCLUSIONS

When sequence variation is appropriate and gene trees are consistent, standard Sanger sequencing of a small number of markers may be all that is required to infer robust and meaningful phylogenetic trees. For species complexes and rapid radiations (either ancient or recent) where this is not the case, the usefulness of sequence datasets will inevitably be limited by the resolution of individual gene trees. Our results suggest that under these circumstances, where the need for NGS and targeted sequence capture, such as hybrid enrichment, is greatest, "made to measure" markers identified using both transcriptome and WGS data of related taxa will deliver results that are superior to those that might be obtained using a more universal "one size fits all" approach. Once available, such markers may nevertheless be useful across a fairly wide range of related taxa: e.g., those presented here, targeted for use in *Erica*, fall within the range of sequence variation that would in principle be applicable across the family Ericaceae. Transcriptome data for many flowering plant groups are now available; these would ideally (but not necessarily) be complemented with WGS or genome skimming data of one or more focal taxa for use in marker selection.

With such data to hand, biologists are still reliant on bioinformatics skills or user-friendly tools (such as MarkerMiner). In either case, the full potential of the techniques will only be harnessed if comparisons to distantly related genomes and generalisations of single/low copy genes across wide taxonomic groups are avoided. We would conclude that rather than searching for ''one size fits all'' universal markers, we should be improving and making more accessible the tools necessary for developing our own ''made to measure'' ones.

## ACKNOWLEDGEMENTS

The authors thank Mark Chase and the 1,000 Plants (1KP) project for access to *Rhododendron* transcriptome data; Kai Hauschulz (Agilent), Abigail Moore (University of Oklahoma), and Frank Blattner, Nadine Bernhardt and Katja Herrmann (IPK Gatersleben) for help and advice with lab protocols; and reviewers Daniel Campo and Ryan Folk and handling editor Keith Crandall for constructive criticism.

### Funding

The work was supported by the Deutsche Forschungsgemeinschaft (DFG; PI1169/1-1 to MDP); South African National Research Foundation (NRF; DUB and MDP). The funders had no role in study design, data collection and analysis, decision to publish, or preparation of the manuscript.

### Grant Disclosures

The following grant information was disclosed by the authors:
Deutsche Forschungsgemeinschaft: PI1169/1-1.
South African National Research Foundation.

### Competing Interests

The authors declare there are no competing interests.

### Author Contributions

- Malvina Kadlec conceived and designed the experiments, performed the experiments, analyzed the data, wrote the paper, prepared figures and/or tables.
- Dirk U. Bellstedt and Nicholas C. Le Maitre contributed reagents/materials/analysis tools, reviewed drafts of the paper.
- Michael D. Pirie conceived and designed the experiments, wrote the paper.

### Field Study Permissions

The following information was supplied relating to field study approvals (i.e., approving body and any reference numbers):

Plant material was collected in the field under permits from Cape Nature and South Africa National Parks.

## DNA Deposition

The following information was supplied regarding the deposition of DNA sequences:

Sequence alignments are available in a Supplemental File.

Raw sequence reads are available from NCBI: PRJNA388814.

## Data Availability

GitHub: https://github.com/MaKadlec/Select-Markers/tree/AllMarkers; https://github.com/MaKadlec/Select-Markers/tree/BestMarkers.py.

## Supplemental Information

Supplemental information for this article can be found online at http://dx.doi.org/10.7717/peerj.3569#supplemental-information.

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
