# Peer review of "Targeted NGS for species level phylogenomics: “made to measure” or “one size fits all”?"

_PeerJ, doi:10.7717/peerj.3569_

## Round 0.1 · original submission · Major Revisions

I have now obtained two thorough reviews of your paper and based on the reviewer recommendations, I am recommending 'major revisions'. The biggest issues seems to be with the claim of using probes for better phylogenetic resolution coupled with an example that simply doesn't support your claim. As reviewer 1 points out, this is problematic. Not sure how you will fix it, but I can think of a few different ways to go (change the example, don't make the claim, etc.). Either way, I think there is substantial work to be done in a revision and would like to send it back to reviewer one for another look. Good luck with your revision.

·

Basic reporting

In this paper, the authors address the question of what approach should be used to develop probes for targeted sequencing efforts in species-level phylogenomics, when there are no genomic data available for the focal taxa. According to the authors, there are two main approaches, one where the probes are designed using available data from more or less distantly related taxa ("one size fits all" approach), and another that calls for an initial investment to generate genomic data in at least two of the species within the studied taxonomic group, and then use this genomic data to design clade-specific probes ("made to measure" approach). To address this question, the authors develop a set of custom Python scripts, and apply their pipeline to an empirical dataset they generate for the flowering plant genus Erica. The authors also compare their approach with two other currently available programs. The main conclusions of the paper are that 1) the "made to measure" approach seems to be better at developing phylogenetically informative sequencing data, and that 2) their custom pipeline proves to be a more powerful tool than the currently available programs (Hyb-Seq and Marker-Miner).

Although the paper is generally well written, there are some parts that are hard to follow and should be re-written to make it clearer and more concise.
Here are some tips for improvement:
- The manuscript is in general too long, and some parts contain an unnecessary amount of information, which makes it sometimes hard to follow. Just to cite a couple of examples, the first three paragraphs of the Introduction can be reduced, and maybe the first two could even be combined into a shorter paragraph. On this respect, there are too many pieces of additional information given in brackets all throughout the text that mean constant interruptions to the reading (for example in the section "In silico comparison with empirical data").
- On the contrary, the last paragraph of the Intro, where the authors describe the aims of the paper, should actually be extended a bit to give a more detailed explanation and justification of what the authors want to accomplish, why and how.

Experimental design

This work presents an interesting novel approach for the generation of phylogenomically informative data, and provides evidence supporting the idea that an initial investment in developing clade-specific probes would pay off in the form of better datasets. This type of work comes in timely given the recent development in NGS technologies, and the increasing number of researchers applying NGS in evolutionary studies.
However, there are several major issues that should be addressed before the paper can be published. Both are related to the use of the genus Erica as the focal/testing taxonomic group.
My main concern comes from the fact that the authors claim their method to be great at designing probes that will be useful to generate phylogenetically informative data, yet they don't seem to be able to resolve the phylogeny of their focal group, the genus Erica. They generate an empirical dataset from different species of this genus to test their approach, and much information is given about the number of probes and target sequences obtained, but very little regarding the resulting phylogenies, which ultimately is the reason for generating such target sequences. From what I understand in figure 6, those trees have been generated using the most variable sequences they obtained, and yet, bootstrap support is very low for most branches, meaning those trees are not well resolved. In the last paragraph of the Discussion, authors argue that long sequences are needed to resolve complex phylogenies, like expectedly that of the Erica group, but it doesn't really seem to work in this case. Doesn't this invalidate their approach? Or at least (strongly) suggests that a different focal taxonomic group should be used to test it?

Along the lines of this, in the paragraph starting in line 474 ("An alternative to optimizing..."), authors mention that their scripts allow for optimization of probe selection based on sequence variability (in addition to length variability), but the lack of data to compare with their ingroup (Erica?) prevents the use of such option in this case. If authors are trying to sell their approach as a more powerful alternative to what is currently available, they should then choose a different focal group, one that allows the use of all the capabilities of their scripts.

In addition, an important component of the paper is the comparison between the authors' custom pipeline for probe design and the two currently available programs (Hyb-Seq and Marker-Miner). An entire section of the discussion is dedicated to this ("The impact of method for marker selection"). However, that is not mentioned as one of the aims of the work. It should be.

Validity of the findings

Please see my comments and concerns in the Experimental Design section.
No further comments added here.

Additional comments

Here is a list of minor edits that authors should consider to improve the manuscript:

- Lines 64-65: RAD stands for "Restriction-Site Associated DNA", not "reduced-...". Please change.
- Lines 76-77: I don't know what that sentence refers to.
- Line 162: please explain here what transcriptome and WGS data is used, i.e. from what species.
- Lines 184-185: The authors should explain in more detail here how their approach differs from Hyb-Seq and MarkerMiner.
- Lines 249-251: I don't quite get this part. Maybe rephrase to make it clearer?
- Lines 258-262: I did not understand this paragraph the first time I read it; I had to read the entire paper, and then read this part again to understand it. Please rewrite.
- Lines 264-279, section "Lab methods": authors go from DNA extraction to library preparation/hybridization enrichment, but they do not explain how they prepare or obtain the probes.
- Line 286: what do authors refer to with "sequence targets"? Is that the transcriptome and WGS data they mentioned in line 162? In general, the terminology is a bit confusing sometimes throughout the manuscript; for example, what is the difference between "selected targets" and "sequence targets"?
- Line 359: It is 132 markers, not 134, isn't it?
- Lines 457-458: I don't understand how selecting "the most appropriate markers" allows the dilution of the baits during the library preparation/hybridization enrichment protocol. Please explain, because the reduction of the protocol's cost seems to be a relatively important claim of the paper.
- Figures 4 and 6 are out of the page range, at least in my printed copy. If this is a format problem, please correct it.

·

Basic reporting

The English is very good and overall the paper is well-written; in a few places there are perhaps some colloquialisms (e.g., the last sentence before Conclusions), but nothing stood out terribly.

There are copious supplemental analyses and data provided. While I will leave it to the discretion of the authors and editors, I personally think more of the empirical data should be disseminated. In your place I have customarily deposited raw NGS data in the Sequence Read Archive. This allows individuals to potentially pull out sequences beyond your capture experiment, or try other assembly methods.

More text should be added to the paper explicitly saying that the alignments, etc., are available as supplemental data; my instinct is to head to Dryad.

Experimental design

I have no concerns about the originality of the paper and its rigor; the methods are quite detailed.

Validity of the findings

I have no concerns here.

Additional comments

This is a very well-written paper on phylogenomic marker development, a topic of continued relevance. In summary of the following comments, I would recommend this for publication with some very trivial revisions.

I very much agree with the underlying message of this paper that "one-size-fits-all" approaches are the wrong direction for many people's projects (yet this is where the community seems to be going in large part, especially on the zoological side), and I think there is still very much a place for marker development tools.

The lab methods are fairly straightforward, and the sample size is very typical for a pilot study of this kind. Please see elsewhere in this review my comments on data availability.

One thing that is not immediately obvious to readers: extensive discussion is devoted to how the intronic variation is more directly addressed here, but it is not immediately clear whether this tool can be used to directly design intronic baits. This might be useful for individuals who are working at very low taxonomic levels; even if flanking region capture is possible, it sets minimum limits on how little sequence in the dataset can be conserved (perhaps no less than half). If introns were directly targeted (in a suitable use case where this is wise and would likely work), the percentage of informative sequence might be much higher. A conspicuous absence has been the lack of automated tools for identifying introns more directly from input WGS data (similar to the Folk et al. paper, but more high-throughput).

One defense of the MarkerMiner approach might be that it is more suited to deep-level phylogenetics. Do you think your approach outperforms it there too? This is hinted at early in the paper, but not developed later.

I did not attempt to run the code (other than making sure it actually executes), but I took a quick look and it appears to be neat, well-documented (other than the need for a readme file on GitHub, which should be fixed upon publication, also I think a toy dataset should be provided for people to troubleshoot with), and show signs of optimization (e.g. parallelization).

The figures are very good.

Minor issues:

497: I believe the citation of Lanier, Huang & Knowles, 2014 is misleading, or perhaps only vague. More variable loci driving an analysis in a problematic way (the way the sentence seems to imply to me) is especially characteristic of concatenation approaches; species tree analyses more equally weight gene partitions by their nature (but are greatly improved by good gene trees).

Fig. 1 -- this is a very complex diagram. Can the text in each box be simplified? Also: vs to vs. and Sricly to Strictly

Given the minimum sequence identity of 73% observed, can you please comment on whether the capture was still effective or whether there was any sequence dropout? Pushing the hybridization to its limits will be of general interest as a short comment, I think.

92-93: Moriarty Lemmon should just be Lemmon

101: On my end there is a corrupted character of some kind in the pdf here and several places afterwards

158: Change "sequence length" to "cumulative sequence length"

171-173: Rewrite to reduce the number of clauses

428: Change "fared poorly" to "resulted in the lowest locus quality" or something more objective

---

## Round 0.2 · accepted · Accept

Thank you for your careful revision accommodating the previous concerns of reviewers. I sent your paper back to one of these reviewers (the harshest) and s/he is now satisfied with your revision as am I. Note this reviewer does make a comment about the inclusion of your phylogeny figure when your focus is on marker development. However, I leave this decision up to you.

·

Basic reporting

In my opinion, the authors have done a great job at dealing with my comments and objections. Though with little change to the original manuscript, I think the paper is now much clearer and easier to follow than it was before.
However, I still have a bit of a concern with the presentation of the results. Please see my comments below, under "Experimental design".

Experimental design

Regarding my main concern about the choice of the focal taxa, the authors responded this:

"This paper is about getting the best possible dataset for difficult phylogenetic problems. There is no guarantee that the results will be conclusive for a given group, but that is beside the point: if you set out to obtain a suboptimal dataset, the chances of success must be lower. We believe our results show clearly that our approach will deliver more, more appropriate, sequence markers.
As correctly inferred by reviewer 2, the data that we present is effectively that of a pilot study. Nevertheless, we could certainly have presented a more flattering representation of the results: we are now presenting more trees, selected on the basis of numbers of supported nodes rather than informative sites."

The authors are right when they say that the main point of the paper is the introduction of their own approach for marker selection, and I recognize now that the dataset they present is what it is, and that there is indeed no guarantee that the results will be conclusive for a given group.
Now, on this respect, I would argue that perhaps the inclusion of the poorly resolved phylogenetic trees as a main figure of the paper (Figure 6) is not really helping their cause. If, like they say (and I now agree), the focus of the paper is on the method for marker selection, perhaps they will be better off by not including the trees as a result (or at least not as a main figure). I think the rest of the results, and the discussion make a great case for their approach, and clearly show that it outperforms the current methods.
I am just concerned that readers will loose focus from the main point of the paper (the marker selection approach) due to the poorly resolved trees, like it happened to me.

In any case, I am not going to hold acceptance of the paper based on this, because it might be as well just me. But if authors agree with me on the above, I would suggest they change it in a final version.

Validity of the findings

No comment.

Additional comments

No comments.